# Designing nanohesives for rapid, universal, and robust hydrogel adhesion

Zhao Pan[1,2,6], Qi-Qi Fu ®[3,6], Mo-Han Wang[4,6], Huai-Ling Gao[1], Liang Dong[1,2], Pu Zhou[4], Dong-Dong Cheng[4], Ying Chen[3], Duo-Hong Zou[4], Jia-Cai He[4], Xue Feng ®[5] ✉ & Shu-Hong Yu ®[1] ✉

Nanoparticles-based glues have recently been shown with substantial potential for hydrogel adhesion. Nevertheless, the transformative advance in hydrogel-based application places great challenges on the rapidity, robustness, and universality of achieving hydrogel adhesion, which are rarely accommodated by existing nanoparticles-based glues. Herein, we design a type of nanohesives based on the modulation of hydrogel mechanics and the surface chemical activation of nanoparticles. The nanohesives can form robust hydrogel adhesion in seconds, to the surface of arbitrary engineering solids and biological tissues without any surface pre-treatments. A representative application of hydrogel machine demonstrates the tough and compliant adhesion between dynamic tissues and sensors via nanohesives, guaranteeing accurate and stable blood flow monitoring in vivo. Combined with their biocompatibility and inherent antimicrobial properties, the nanohesives provide a promising strategy in the field of hydrogel based engineering.

Living systems are characterized by a diverse range of adhesion in wet environments. From the adhesion of cells to extracellular matrix, tendons and ligaments to bone[1], to mussel threads to rocks[2], the adhesion of biological tissues to each other or to other solids is considered as a fundamental mechanism to support the structural integrity and functioning of organisms. Recently, in attempts to imitate the perfection of nature and meet the challenges in the field of biomedical engineering, such as the nascent field of hydrogel machine[3], intense efforts have been dedicated to achieving robust wet adhesion between hydrogels and various solid materials[4]. Researchers have designed a variety of wet adhesion approaches, ranging from adhesion strategies inspired by natural organisms, such as mussels[5], sandcastle worms[6], slugs[7], or barnacles[8–10], to surface-initiated[11] or ultrasound-mediated adhesion building techniques[12]. The possibility for interfacing and integrating hydrogel with other solids, has offered the opportunities to

fuel transformative advances in numerous technologies across a wide range of applications. For instance, a microgel coating on a PVC substrate achieved robust and anticoagulant implantable biomedical devices[13]. The timescale-dependent adhesion enabled by the electrical oxidation approach facilitated the fault-tolerance of hydrogel surgical tapes[14]. A pulsatile releasing platform based hydrogel adhesion and microcapsule techniques presented a promising strategy for scarless skin wound repair[15].

Nanoparticles have been demonstrated in a sequence of studies to work as interfacial interlinks, bonding hydrogels or biological tissues together[16–18]. Then, a range of nanoparticles-based glues with diverse physicochemical properties associated with the nanoparticles has previously been explored. For example, mesoporous silica nanoparticles (MSN) glues contributed to the activation of benign inflammation[19]; Ceria decoration endowed MSN glues with

[1]Department of Chemistry, New Cornerstone Science Laboratory, Institute of Biomimetic Materials & Chemistry, Anhui Engineering Laboratory of Biomimetic Materials, Division of Nanomaterials & Chemistry, Hefei National Research Center for Physical Sciences at the Microscale, University of Science and Technology of China, Hefei 230026, China. [2]Zhejiang Cancer Hospital, Hangzhou Institute of Medicine (HIM), Chinese Academy of Sciences, Hangzhou, Zhejiang 310022, China. [3]Institute of Flexible Electronics Technology of THU, Jiaxing, Zhejiang 314000, China. [4]Department of Oral Implant, Stomatology Hospital & College, Anhui Medical University, Key Laboratory of Oral Diseases Research of Anhui Province, Hefei 230026, China. [5]AML, Department of Engineering Mechanics, Centre for Flexible Electronics Technology, Tsinghua University, Beijing 100084, China. [6]These authors contributed equally: Zhao Pan, Qi-Qi Fu, Mo-Han Wang. ✉e-mail: fengxue@tsinghua.edu.cn; shyu@ustc.edu.cn

ROS-scavenging activity[20]; silica nanoparticles glues promoted blood coagulation[21]; metal oxide nanoparticles glues could provide high contrast effects for real-time imaging[22], or possessed antimicrobial properties[23]. In tandem with enhancing versatility, researchers were also concerned with improving the adhesive properties of nanoparticles-based glue. Modulating both the surface area of nanoparticles and their assembly form could increase the contact area between nanoparticles and hydrogel polymer chains, enhancing the cohesion between nanoparticles and contributing significantly to the adhesion performance[24,25]. In addition, both experimental and theoretical efforts have proven that the shape of nanoparticles has a substantial impact on adhesion enhancement[26,27]. However, the adhesion energy supplied by the developed nanoparticles-based glues mostly remains below 100 J/m², significantly lower than the biological adhesion (e.g., tendon-bone adhesion, ~800 J/m²). Furthermore, the range of their adherends was limited to specific hydrogels and biological tissues. Similarly to other glue-based hydrogel adhesion approaches[28–30], nanoparticles-based glues typically necessitate more than a few minutes for adhesion formation, unlike the rapidity of reported tapes and patches that require only a few seconds[31]. Considering the surface chemistry of nanoparticles and the inherent properties of hydrogels, such as mechanics and water absorption capability, may profoundly influence the performance of adhesives. In light of this, we postulate that modulation and optimization of these factors could enhance the adhesion performance of nanoparticles-based glues, thereby promoting their practicality and application potential.

Herein, we describe the design of a type of adhesives, namely nanohesives, comprising a surface-activated nanoparticles (ANP)-based glue and a matched dissipative hydrogel (Fig. 1a). We attempt to reinforce the performance of hydrogel adhesion achieved by nanohesives, denoted as nanohesion, by modulating both the composition of dissipative hydrogel and the surface chemistry of the nanoparticles.

The designed dissipative hydrogel could rapidly absorb the water, organize the ANP at the interface, and establish nanohesion through ANP interlinking within seconds. Owing to the robust and universal interactions between ANP to the dissipative hydrogel and solid surfaces in wet environments, the nanohesives exhibit broad-spectrum adhesion capacity. Moreover, the dissipative mechanics of the hydrogel effectively resist damage to the adhesive interface, thus toughening the nanohesion.

## Results

### Design and preparation of nanohesives

To implement the proposed design of the nanohesives, the initial step involved the preparation of surface activated nanoparticles (ANP) by silanization modification of pristine silica nanoparticles (Supplementary Fig. 1). The resultant ANP are modified with carboxylic groups on the surface, without changing the size of nanoparticles while providing abundant adhesive sites. The matched dissipative hydrogel was prepared by mixing two types of crosslinked polymer networks, which are the long-chain covalently crosslinked network and physically crosslinked agarose network. To enhance the interaction with ANP, positively charged groups with quaternary ammonium salt (10–50 mol%) were incorporated into the long-chain network of polyacrylamide. In addition, the folding structures within the agarose network act as weak sacrificial bonds to promote the energy dissipative capacity of the hydrogel and resist the fracture break of the nanohesion (Supplementary Fig. 2a).

The dissipative hydrogel can be engineered into single-sided or double-sided tapes owing to its high manufacturing flexibility (Supplementary Fig. 2b, c). The ANP were applied to the interface in the form of an aqueous dispersion (20 wt%) in favor of wetting the interfacial surfaces. The applying procedure is convenient by simply dipping, brushing, or spraying ANP glue onto the interfaces, then attaching hydrogel tape upon the dispersion (Fig. 1b). Due to the

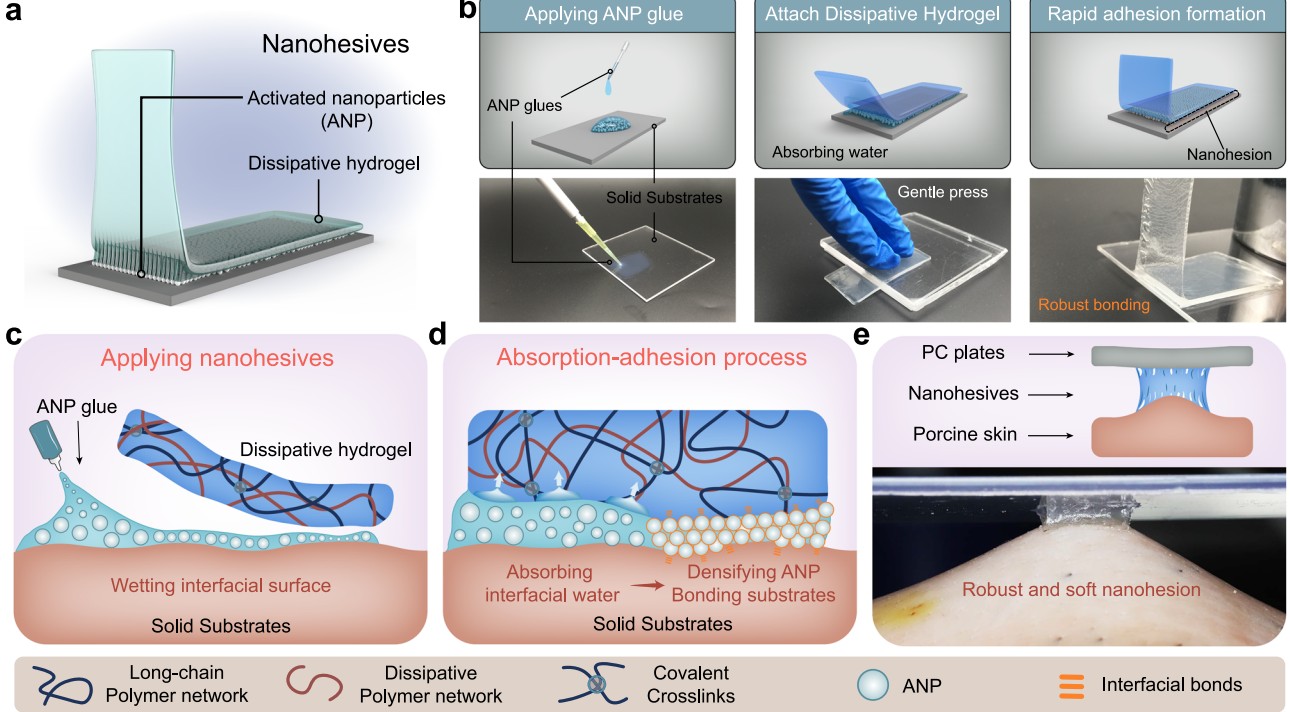

**Fig. 1 | Design and preparation of nanohesives. a** Schematic illustration for the design of nanohesives. The nanohesives comprise a layer of activated nanoparticles (ANP) and a dissipative hydrogel. **b, c** The nanohesives were achieved conveniently by applying ANP glue to the interface, then attaching the dissipative hydrogel above. **d** The absorption of the interfacial water by dissipative hydrogel densifies the ANP and facilitates the formation of nanohesion. **e** A robust and soft nanohesion achieved between biological tissues (represented by porcine skin) and engineering materials (represented by polycarbonate, PC plate), could withstand large deformation.

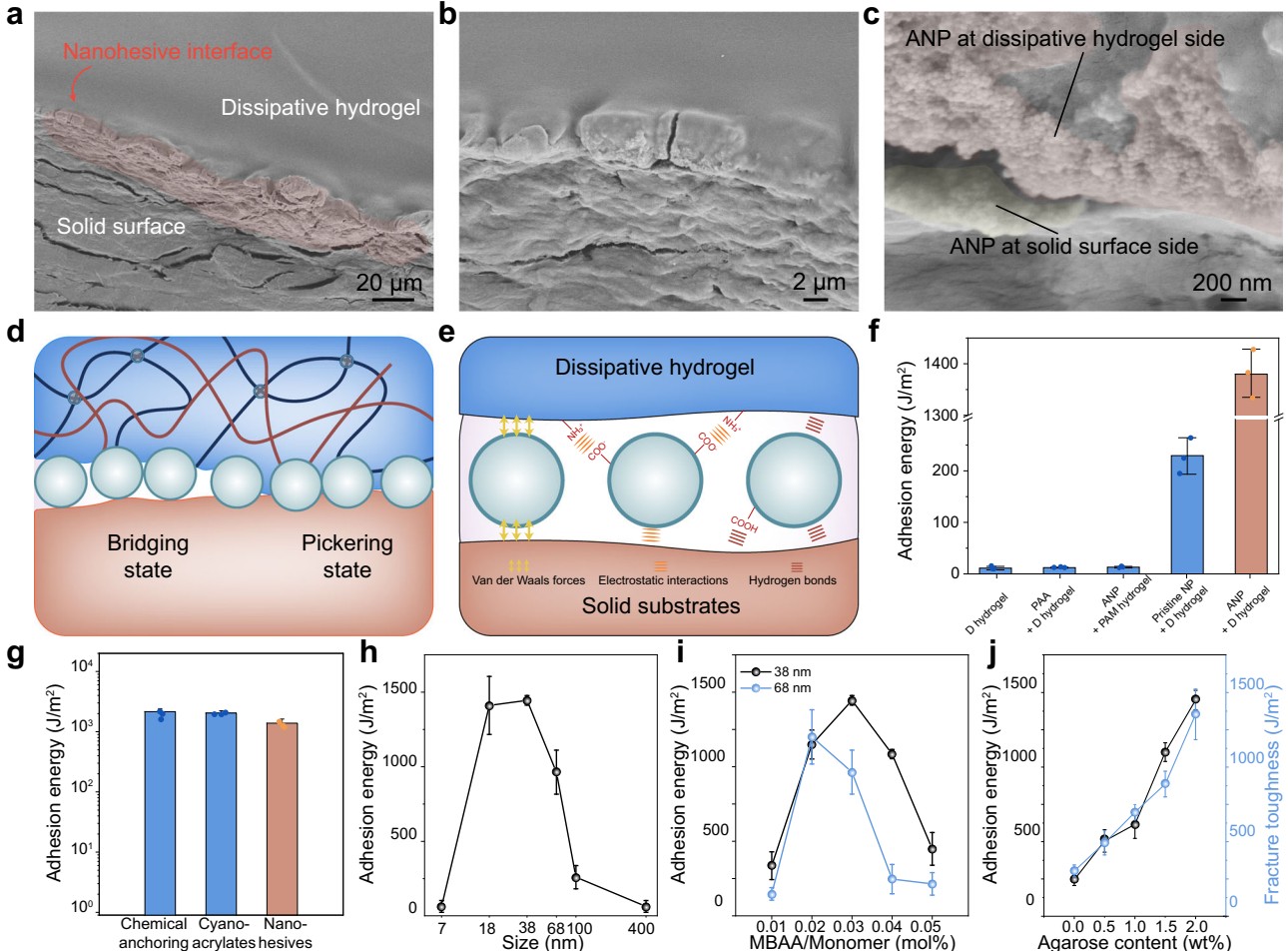

**Fig. 2 | Adhesion mechanism of nanohesives. a** SEM analysis of adhesive interface in nanohesion. **b** Clear morphological distinction is shown at adhesive interface. Magnified images in (**c**) reveal that layers of ANP bridge the surfaces. SEM observation was repeated three times independently, yielding similar results. **d** Two states of ANP interlinking, the bridging state and Pickering state, are presented at the interface. **e** The interactions between ANP and surfaces were the collective short-range forces. **f** Evaluation of adhesion between nanohesives and substrates with varied nanoparticles-hydrogel interactions. D hydrogel represents dissipative hydrogel, PAA represents polyacrylic acid, and PAM represents polyacrylamide. **g** Comparison of nanohesion (bonding hydrogel to pristine glass) versus chemical anchoring (between hydrogel network and vinyl modified glass surface) and cyanoacrylates (bonding hydrogel to PC plate). **h** Adhesion energy of nanohesives with different ANP sizes. **i** Adhesion energy of nanohesives with two different ANP sizes, to hydrogels with varying crosslinking degree. **j** Adhesion energy and matrix toughness vary with the energy dissipative agarose content in hydrogel. Values in **f–j** represent mean ± standard deviation (s.d.) (*n* = 3 independent samples).

interactions are mostly formed instantaneously after the intimate contact occurs between nanoparticles and solid surfaces, the rate at which nanohesion forms becomes highly dependent on the absorption of interfacial water, which leads to the condensing of ANP at the interface (Fig. 1c, d). Considering that the rate of the interfacial water absorption by the hydrogel depends on the swelling of the hydrogel[31], the incorporation of positively charged groups in a long-chain polymer network could increase the swelling of hydrogel, thus accelerating the absorption rate of interfacial water by the dissipative hydrogel. As a result, a double-sided tape can effectively establish nanohesion, linking engineering solids (represented by polycarbonate plate, PC) and biological tissues (represented by porcine skin), within 3 s (Fig. 1e). Profiting from the softness and toughness of the nanohesives, the fixation is compliant and able to endure large deformation (Supplementary Movie 1).

## Adhesion mechanism of nanohesives

Next, we further investigated the factors affecting nanohesion. Morphological analysis revealed that the nanohesion interface constitutes a distinct sandwich structure, and ascertains that a layer of ANP bridges the dissipative hydrogel and the adherends (Fig. 2 a–c). Parts

of the aggregated ANP are trapped in the hydrogel, while the other tips are in contact with the adherent substrate. Previous theoretical studies predicted that nanoparticles interlink surfaces at interfaces mainly with two models[18]. They may form a bridge connecting two surfaces together, namely the bridging state; or partition between two soft substrates to form interlinks, namely the Pickering state (Fig. 2d). According to well-established facts in nanoscience, the interactions between individual nanoparticles and each other or substrate surface in water, are collective effects of short-range forces of primary electrostatic interactions, van der Waals forces, and hydrogen bonds[32]. These short-range forces are the basis for the broad-range adhesion capability of nanohesives (Fig. 2e). The nanohesion formed by physical interaction accumulation provides adhesion energy comparable to that provided by the covalent interlinks formed through chemical reaction and the glassy layer adhesion polymerized from cyanoacrylates (Fig. 2g).

Since the interfacial energy of nanohesion is mainly attributed to the dissipative hydrogels, the strength of interaction between ANP and hydrogel is expected to affect the interfacial toughness of nanohesion. First, the affinity of nanoparticles to hydrogels is typically influenced by the size of the nanoparticle and gel mesh.

When the size correspondence between ANP and hydrogel is adjusted, the interaction between them is supposed to be influenced. The results revealed a non-monotonic behavior of adhesion energy of nanohesion versus the size of nanoparticles (Fig. 2h). The variation in the mesh size of hydrogel, which regulated by the content of crosslinking agent, corresponds to different peak adhesion energy values for two different sizes of ANP, validating that the strength of interfacial interaction influences the adhesion energy of nanohesives (Fig. 2i). The size characterization of nanoparticles is presented in Supplementary Fig. 3.

In addition, the influence of interface chemistry on adhesion was investigated (Fig. 2f and Supplementary Movie 2). When the dissipative hydrogel was directly attached to the substrate without ANP, the resulting adhesion energy measured around $10 J/m^2$. The adhesion energy was also quite low ($10 J/m^2$) if a solution of polyacrylic acid, which carries a high amount of carboxyl groups like ANP, is used as the glue. In previous reports, polyelectrolyte polymers were employed as stitching molecules to mediate the tough adhesion of hydrogel to substrate[28–30]. The low adhesion energy measured here might be because of the use of an impermeable solid as the test substrate, and the lack of chemical anchoring designed between the stitching molecule and the test substrate. Consequently, the effective formation of a typical topological adhesion (typically beyond $300 J/m^2$) between the hydrogel and substrates was hindered. However, even when ANP glue was used to mediate adhesion, pristine polyacrylamide (PAM) hydrogel exhibited limited adhesion with substrates if the designed positive charged groups were absent in the hydrogel network. The electrostatic interaction facilitated by carboxylate-quaternary ammonium bonds strongly reinforced the interfacial adhesion. Through surface activation, ANP featuring carboxyl groups increased the adhesion energy from $250 J/m^2$ to $1500 J/m^2$, compared to the pristine nanoparticles with silanol groups. These results validated that the breadth and strength of nanohesion were substantially enhanced by interfacial chemical design.

Furthermore, the interfacial adhesion energy of hydrogel bonding in nanohesives is predominantly influenced by the dissipative capability of the hydrogel. To investigate this influence, hydrogels with varying agarose contents were prepared, establishing a dissipative network. The fracture energy of these hydrogels increased with higher agarose contents, indicating an augmented dissipative capability (Fig. 2j). Additionally, the presence of MOTAC in the hydrogel network improved adhesion but concurrently decreased mechanical performance. The overall adhesive energy exhibited a non-monotonic variation with increasing MOTAC content, reflecting the combined effects of fracture toughness and bonding sites (Supplementary Fig. 4). Compared to the adhesion energy reported in previous studies on nanoparticles-based glues ($<100 J/m^2$), the adoption of the dissipation mechanism in hydrogel led to a significant increase of adhesion energy, even when the similar pristine nanoparticles were employed (Fig. 2f). The energy dissipation capacity of hydrogel mainly relies on the hysteresis effect brought by agarose network, which was verified by the linear correspondence between agarose content and the fracture toughness of hydrogel (measured via pure-shear test). The linear correspondence between agarose content and nanohesion energy then indicates that the enhancement of hydrogel fracture toughness increases the resistance to damage of the nanohesion interface. A comprehensive comparison between ANP and previously reported nanoparticle-based glues reveals the nanohesives presented in this study outperform the reported adhesive performance on the widely used PDMA (Poly N,N-dimethylacrylamide) hydrogel (Supplementary Table 1).

We propose that the mechanism underpinning the formation of nanohesion involves an absorption-interaction process. In this study, the introduction of charged functional groups into the hydrogel matrix significantly enhances its water-absorption rate. The dissipative hydrogel rapidly absorbs the interfacial water layers from both the adhesive and the substrate surface. Consequently, the density of dispersed nanoparticle aggregates in the ANP glue increases due to dehydration, facilitating their intimate contact with the substrate. The absorption-interaction mechanism driving nanohesion formation facilitates rapid bonding, as evidenced by our experimental tests (Supplementary Fig. 5a). While higher concentrations may potentially result in faster bonding speeds, the practical significance of such acceleration might be challenging to discern. Conversely, when using lower concentrations of the dispersion, the adhesive energy tends to decrease (Supplementary Fig. 5b). We attribute this effect to a reduction in the distribution density of nanoparticles at the interface as the concentration decreases, thereby diminishing the number of contact points for interfacial adhesion and consequently reducing the intrinsic bonding toughness.

## Universal and tough nanohesion

The universality of nanohesives could facilitate the enhancement of flexibility and convenience of interface design. To evaluate universality, we examined the adhesion of nanohesives to a wide variety of engineering solids with different compositions and surface properties, as well as biological tissues.

Engineering solids that undergo no surface chemical treatment commonly appear to feature inert surfaces. In order to build hydrogel adhesion with such engineered solids, various subtle methods have been developed to pre-treat the substrate surface so that it could form covalent bonds with the hydrogel. For instance, in order to establish hydrogel adhesion, polydimethylsiloxane (PDMS) was required to modify through silanization, while polyimide and polycarbonate were functionalized with primary amines[31]. The establishment of stable wet adhesion on various engineering solid surfaces with one versatile adhesion strategy would facilitate the accessibility and practicality of hydrogel adhesion but remains a great challenge. The establishment of nanohesion is based on the rapid formation of widely available short-range forces between small-sized nanoparticles and surfaces. Benefiting from this mechanism, nanohesives can implement robust adhesion to almost all surfaces of non-pre-treated engineering materials, ranging from metals ($-1400 J/m^2$, Supplementary Movie 3), ceramics ($-1400 J/m^2$, Supplementary Movie 4), plastics ($500–1400 J/m^2$, Supplementary Movie 5) to rubbers ($500–1350 J/m^2$, Supplementary Movie 6) (Fig. 3a). Among these materials, polytetrafluoroethylene (PTFE) is notably anti-adhesive and is usually fixated by mechanical fastening or passive embedding in practice. However, the adhesion energy of nanohesives to PTFE is found to be around $500 J/m^2$, which is much higher than that achieved by commercial Scotch tape or cyanoacrylates glue (Fig. 3b and Supplementary Fig. 6). This performance could prove beneficial in attaching PTFE to surfaces in practice to facilitate the development of applications such as biomedical device assembly or tissue separation. In addition, irregular geometries, such as grooves, wells, crevices, and appendages on substrate surfaces, often impede the effectiveness of adhesives, especially those adhesive patches based on surface geometric designs. Hence, to investigate the adhesion performance of nanohesives to irregular surface, a group of contrast experiments was implemented on polylactic acid (PLA) membranes with sandpaper and a silicon wafer as templates, respectively (Fig. 3c and Supplementary Fig. 7). Attributing to the fine size and the favorable wetting behavior of ANP, nanohesives exhibited great tolerance to surface roughness.

The establishment of robust wet adhesion to both engineering solids and biological tissues, is of vital importance for medical applications such as tissue repair, tissue engineering, and drug delivery. It is regarded as an imperative technology for addressing the human-machine interface challenges in the future[33]. Therefore, we further

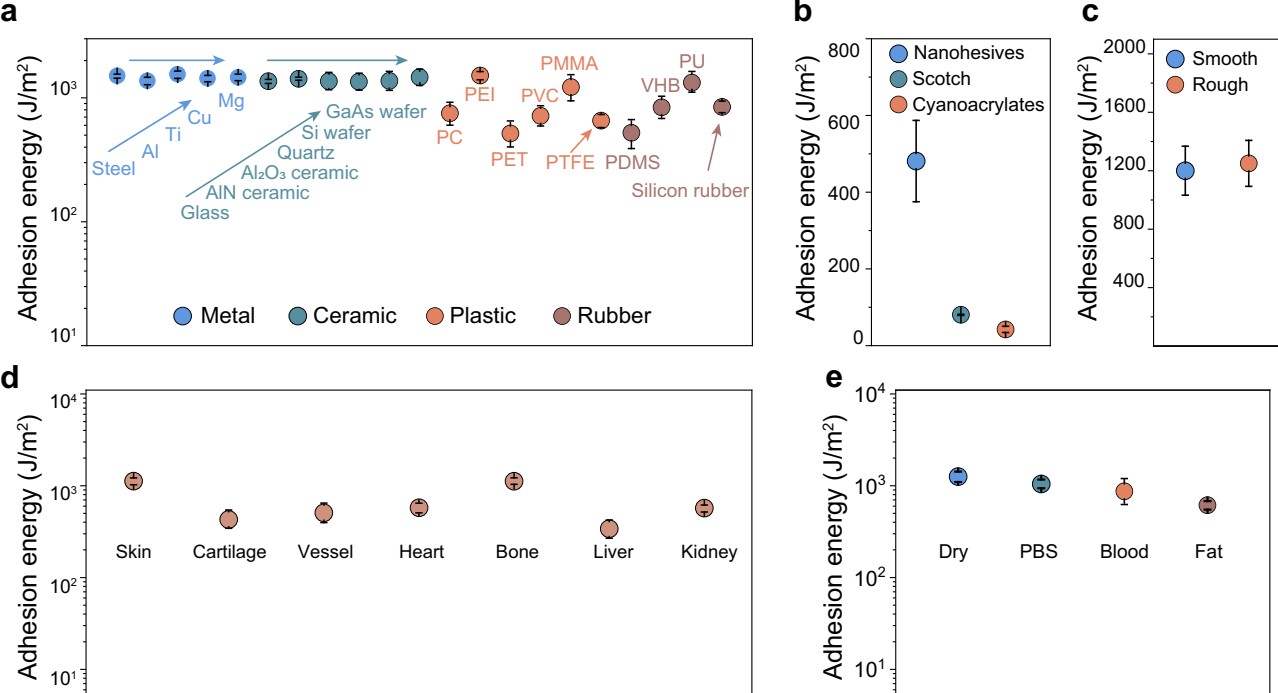

**Fig. 3 | Universal and tough nanohesion. a** Adhesion performance of nanohesives to engineering materials including metals, ceramics, plastics and rubbers. **b** Comparison of nanohesives, Scotch and cyanoacrylates (CA) adhered to PTFE. **c** Adhesion energy of nanohesives on PLA surface with different surface roughness. **d** Adhesion to various biological tissues. **e** The tolerance evaluation of nanohesives to various body fluid foulants was test on porcine skin. Values present mean ± standard deviation (s.d.) ($n$ = 3 independent samples).

investigated the adhesive performance of nanohesives towards various biological tissues ex vivo.

To the biological tissues such as skin (1200 J/m²) (Supplementary Movie 7) and bone (1200 J/m²), the nanohesion energy was high since those tissues are comparatively robust and could sustain structural integrity under deformation during the test until the interface was peeled (Supplementary Fig. 8). As for the fragile tissues such as liver (350 J/m²) and kidney (600 J/m²), significant decline in adhesion energy occurred because the surface of those tissues was ruptured before the nanohesion failing during the peeling-off process (Fig. 3d). Generally, the establishment of adhesion inside the body is quite difficult due to the exposure of adhesive interface to various foulants, for instance, blood or fat. However, nanohesives, being glue-based adhesive family comparable to previous report[7,9], exhibit the capability to exclude the fluid foulants from the adhesive interface under gentle pressure. Subsequently, the rapid formation of nanohesion would protect the interface against being re-fouled (Supplementary Movie 8). When nanohesives were subjected to the surface of tissues covered with phosphate buffer saline (PBS), the adhesion energy was found to be around 1100 J/m², indicating that interfacial water has little impact on the adhesion (Fig. 3e). A similar phenomenon was observed when adhering to blood-covered surfaces (Supplementary Fig. 6, Supplementary Fig. 9, and Movie 9), as well as fat-covered surface (fat side of porcine skin, Supplementary Movie 10).

All these encouraging results about adhesion capability illustrate the potential of nanohesives for practical application, especially in the biomedical field. Nevertheless, their reliability of adhesion in wet environments and their bioactivity remain to be investigated. As the results indicated, the adhesion energy of nanohesives to adherends decline slightly after soaking in PBS after 24 h (1200 J/m² on glass, 1000 J/m² on porcine skin after soakage) (Supplementary Fig. 10). Besides, the nanohesion could keep steady in PBS under a 6 kPa load for at least 24 h. The biocompatibility of nanohesives was investigated in vitro; after being cultured in nanohesives-conditioned medium for

24 h, the human gingival fibroblast cells were able to maintain full viability (Supplementary Fig. 11). Since quaternary ammonium components are widely reported to possess antibacterial activity, we further inspected the in-vitro growth inhibition capability of nanohesives against the model Gram–positive bacteria, S. aureus. The results demonstrated that, compared to the blank group that cultured in normal solid culture medium (SCM) and the control group that co-cultured with dissipative hydrogel without quaternary ammonium components, nanohesives illustrated a significant growth inhibition capability against bacteria.

## Application of blood monitoring through nanohesives-enabled hydrogel machine

Vessel patency monitoring holds significant importance for patients recovering after reconstructive surgeries such as vessel anastomosis; however, clinical monitoring methods are commonly terminated upon patient discharge, leading to a possible loss of the opportunity to timely salvage the ischemic reconstructed tissues. Therefore, biomedical electronic devices capable of directly monitoring blood flow, such as strain sensors, are highly desired for continuous monitoring of vascular patency after hospital discharge. The key criterion for accurate signal monitoring using strain sensors involves reliable fixation onto the target object, but conventional loose fixation methods such as cuff structures obscure the transmission of the strain signal from the vessel to the sensor[34,35]. Aiming to demonstrate the practical application of the nanohesives, a strain sensor is fixated around a blood vessel by the soft adhesive fixation of nanohesives, to real-timely and continuously monitor blood flow in animal trials (Fig. 4).

The strain sensor consists of two sensitive Au resistor strain gauges that connected each other with serpentine traces for conformal contact. After tight conformal adhesive fixation, it presents reliable tensile strain reactivity (Fig. 4a). The simulated signals detected by the stain sensor fixed on nanohesives, showed a strong linear relationship with high repeatability and stability over the strain range

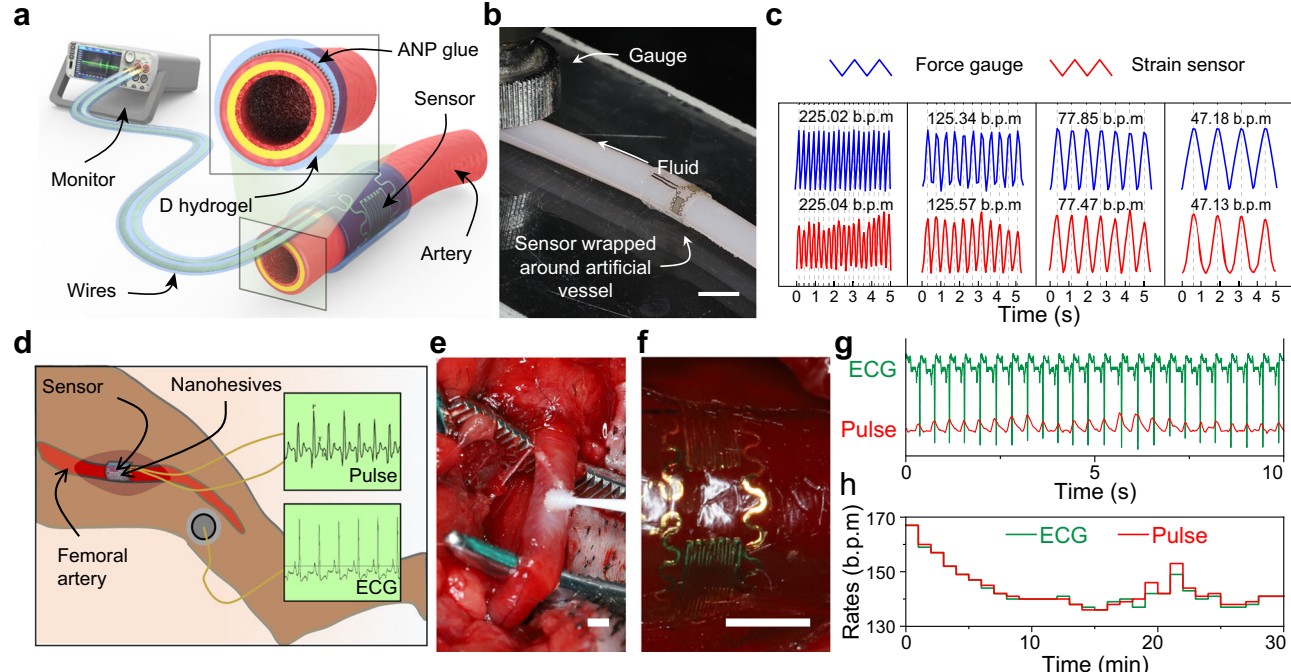

**Fig. 4 | Application of blood monitoring. a** Schematic diagram of the arterial pulsation monitoring set-up. D hydrogel represents dissipative hydrogel. **b** Photograph of the in vitro simulation test set-up, in which the strain sensor is fixed by nanohesives around an artificial artery model. Scale bar: 5 mm. The set-up mimics the pulsatile behavior and typical expansion of blood vessel. **c** Expansion signals in different frequencies simultaneously obtained from force gauge and strain sensor. Attributing to the robust and conformal fixation of strain sensor via soft nanohesives, the peaks between two sets of expansion signals are well matched, and the frequency error is less than 5‰. **d** Schematic diagram of the in vivo monitoring of blood flow on canine femoral artery. **e, f** The in vivo fixation procedure of strain sensor. ANP glue was beforehand brushed on the artery and strain sensor, and the strain sensor was picked up by forming adhesion with dissipative hydrogel. Then, the nanohesives adhered strain sensor was attached around the coating area. Scale bar: 2 mm. **g** The in vivo measured pulse signals via nanohesives-fixed strain sensor have a close correspondence with the simultaneous acquired ECG signals. **h** The fluctuation and comparison of pulse rates that respectively obtained from ECG monitor and nanohesives-fixed strain sensor in a 30-min test.

of blood vessel expansion (typically less than 5%), guaranteeing the feasibility of blood flow monitoring (Supplementary Fig. 12).

To demonstrate the accuracy of monitoring from this strategy, an in vitro simulation experiment is performed on a test set-up of mimicking pulsatile behavior and typical expansion of blood vessels (Fig. 4b and Supplementary Fig. 13). The strain sensor was wrapped around and adhered on the artificial artery (silicon rubber, diameter: 4 mm) by nanohesives. The tube was sealed at one end and connected with a syringe pump at the other. Using the regularly reciprocating pump, a pulsatile behavior was mimicked and the artificial artery was forced to periodically expand in diameter, which can be simultaneously recorded as resistance variation and strain signals from the sensor and the force gauge, respectively. As shown in Fig. 4c, despite the frequencies varying from 47 to 225 beats per minute (b.p.m), which cover the normal heart rate range of human, the frequency error of the data detected by nanohesives-fixed strain sensor is typically less than 5 ‰ when compared with that by force gauge. The high-level compatibility of the peak strain and resistance signals proves the accuracy of this monitoring strategy.

For in vivo pulse monitoring, the canine femoral artery was chosen to carry out the experiment (Fig. 4d). According to the diameter of the femoral artery, the serpentine traces of the strain sensor were specially designed with a lateral length of 7 mm to enable the two strain gauges at the opposite position on the vessel, and the sensor was fixed by nanohesives in the same way as the in vitro test (Fig. 4e, f). After the implantation, the sensor was connected to a digital multimeter to acquire the pulse signals. Meanwhile, an electrocardiograph (ECG) monitor was applied to simultaneously record the rate data from heartbeats as the control for characterizing the precision of this method. The results showed distinguishable and regular in vivo pulse signals obtained by nanohesives-fixed strain sensor, and they match up precisely with the ECG diagram (Fig. 4g). The conformal deformation of nanohesives is the crucial factor for precise in vivo monitoring, because it not only firmly fixates the strain sensor, but also allows the synchronous expansion of the strain sensor and the vessel. A 30 min monitoring was implemented to verify the in vivo stability of nanohesives-fixed strain sensor. The pulse rates obtained from the ECG monitor and nanohesives-fixed strain sensor are respectively counted per minute. The comparison of fluctuations of each data (Fig. 4h) showed that the rate difference is less than 2 b.p.m (beats per minute) in more than 90 percent of the time (Supplementary Fig. 14), and no unauthentic data with a big deviation is recorded. Thus, this showcase application demonstrated that nanohesives have reliable and soft adhesive capability, to conformally and robustly fix engineering devices to animal tissues and permit them to perform the accurate and stable function in vivo.

## Discussion

The majority of theoretical and experimental studies on nanoparticles-based glues have been conducted with pristine chemical surfaces of nanoparticles and single-network hydrogels. This hydrogel adhesion technique is predicated on the intrinsic affinity between nanoparticles and polymer network, make it independent of chemical and physical triggers such as initiators, photo-radiation, heat, and so on. It is a type of highly distinctive wet adhesion from polymer binders and empowers the exploration of diverse and versatile interfacial bonding designs. However, with the rapid development of wet adhesion technology, despite the significant research pertaining to nanoparticles-based

glues in hydrogel adhesion, low adhesion energy, narrow adhesion spectrum and slow adhesion formation rates are ongoing technical challenges that are substantially affecting the efficiency and practicality of nano-based adhesives.

In this article, we report a type of nanohesives that incorporates chemically active nanoparticles and dissipative hydrogel to achieve a practical nanoparticle-based adhesive. These ready-to-use soft nanohesives synergistically combine the broad-spectrum binding capability of nanoparticles, with the energy dissipative capacity of hydrogels. The nanohesives are able to rapidly and robustly adhere to a wide variety of engineering materials and biological tissues, and are tolerant to surface fluid foulants. Although the underlying basic binding mechanism of the nanohesives is analogous to that of previously developed nanoparticles-based glues, their distinctive feature lies in their universal and robust adhesion to engineering solid materials beyond biological tissues. Furthermore, their rapid formation hold the potential to enhance design feasibility and broaden application horizons, representing a significant technological advancement.

We showcased a device implantation application that enables real-time detection of bio-signals, through the rapid and conformal fixation of flexible bioelectronics to dynamic biological tissues using-nanohesives. Representative in vitro and in vivo experiments revealed that the nanohesives are capable of incorporating and functionalizing the shelf-ready implanted device in a physiological environment with biological tissues in a straightforward manner. The demonstrated convenience, non-invasiveness, and stability of this functionality over traditional mechanical fixation methods are in accordance with the academic development of future human-machine interfaces.

Differing from the prevalent polymer-based adhesives, nano-particles can encompass organic, inorganic, or complexes. Moreover, nanoparticles usually exbihit various responsive properties, including but not limited to the nano-enzyme effect, photo-thermal effect, magneto-thermal effect, pro-healing and pro-regeneration benefits, pro-coagulation capacity, controlled release and slow release abilities, ROS elimination property, autophagy induction, apoptosis effect, and so forth. Therefore, nanohesives offer a dual advantage by not only solving the interfacial adhesion problem, but also subtly implementing unique synergistic features on the interface to realize the functional integration of materials and devices.

However, future studies are required to further enhance the adhesion performance of nanohesives and validate their potential in practical applications. While spherical nanoparticles have demonstrated excellent adhesion properties, other shapes of nanoparticles may be able to bring about denser interfacial stacking with lower concentrations, such as nanoplates. Physical interactions can facilitate the rapid bonding of nanohesives to the substrate, but in several environments such as physiological body fluids, the incorporation of covalent bonding mechanisms in the surface chemistry of nanoparticles may facilitate subsequent adhesion stabilization after the initial adhesion is established by physical interactions. However, at the early stages, nanohesives offer a promising alternative for achieving rapid, universal and tough wet adhesion to diverse materials. We envision that the nanohesives furnish valuable insights for future design and exploitation of nanomaterial-based adhesives.

## Methods

### Materials
All the chemicals were used as received without further purification. For the positively charged hydrogels, acrylamide (AAm; Sigma-Aldrich A8887, ≥99%) and [2-(Methacryloyloxy) ethyl] trimethylammonium chloride (MOTAC; Sigma-Aldrich 408107, 75 wt% in $H_2O$) were the monomers used for the long-chain networks, $N,N$-methylenebisacrylamide (MBAA; Sigma-Aldrich 146072, 99%) was used as

the crosslinking agent, 2-hydroxy-2-methylpropiophenone (Irgacure 1173; Sigma-Aldrich 405655, 97%) was used as the photoinitiator. Agarose (Aladdin A104062, for biochemistry) was used for the dissipative polymer networks. For chemical modification of solid materials to build covalent interfacial adhesion, functional silane 3-(trimethoxysilyl) propyl methacrylate (TMSPMA; Sigma-Aldrich 440159, 98%) was used. Silica nanoparticles were synthesized according to the Stöber method. Tetraethylorthosilicate (TEOS; Sigma-Aldrich 131903, 98%) and ammonium hydroxide (Sinopharm Chemical Reagent Co. Ltd. China, 99%) were used. For carboxylic modification of nanoparticles, (3-aminopropyl) triethoxysilane (APTES, Sigma-Aldrich 440140, 99%), succinic anhydride (Sigma-Aldrich 239690, ≥99%), $N,N$-dimethylformamide (DMF; Sigma-Aldrich 227056, 99.8%) and 4-dimethylaminopyridine (DMAP; Aladdin D109207, 99%) were used.

In the adhesion energy test, polycarbonate film (180 μm) was used as stiff backing adhered to hydrogel by ethyl cyanoacrylate (Loctite 403, Henkel). Poly (acrylic acid) solution with an average Mw~100,000 (Sigma-Aldrich 523925, 35 wt% in $H_2O$) was used. Steel plates, aluminum plates, titanium plates, and copper plates were used as metal substrates. Borosilicate glass, aluminum nitride ceramic, aluminum oxide, quartz, silicon wafer, and gallium arsenide wafer were used as ceramic substrates. Polycarbonate (PC), polyetherimide (PEI), polyethylene glycol terephthalate (PET), Polyvinyl chloride (PVC), polymethyl methacrylate (PMMA), and polytetrafluoroethylene were used as plastic substrates. VHB (3M 4910), polyurethane (PU; Smooth-On), and silicone rubber were used as rubber substrates. Porcine skin, liver, heart, kidney, bone, vessel, and cartilage were purchased from a local grocery store. Polylactic acid (PLA) particles (4032D, Natureworks) were used for fabricating PLA films.

Animal studies were in full compliance with the guidelines of the Institutional Animal Care and Use Committee of Anhui Medical University, China (no. LLSC20150002). A 27 kg labrador dog (female, 8 months old) purchased from Nanjing Anlimo was housed in the animal experimental center of Anhui Medical University under standard feeding conditions during the experimental period.

### Synthesis of hydrogels
The hybrid dissipative hydrogel was synthesized by mixing Agarose (250 mg), AAm (2.1768 g), MOTAC solution (3.635 g), Irgacure 1173 (71.25 mg), MBAA (102.5 μL of 2% solution) and $H_2O$ (11.49 g) in a glass bottle. After three degassing cycles, the bottle was sealed under $N_2$ and heated at 97 °C in the water bath. The agarose powder was dissolved and the solution turned transparent after 20 min. Then, the resulting solution was injected into a mold to cool down at room temperature until agarose gel formed. After that, the mold was exposed to UV light (365 nm wavelength, 8W) for 1h to synthesize hydrogel. Through a similar route, the hybrid agarose-polyacrylamide hydrogel was synthesized by mixing agarose (250 mg), AAm (3.032 g), Irgacure 1173 (71.25 mg), MBAA (102.5 μL of 2% solution) and $H_2O$ (11.49 g). The PDMA hydrogel was synthesized by mixing DMA (3.465 g), MBAA (81 μL of 2% solution), Irgacure 1173 (57 mg), and $H_2O$ (10 g) in a glass bottle. After three degassing cycles, the bottle was sealed under $N_2$, and the resulting solution was injected into the mold, and then exposed to UV light (365 nm wavelength, 8W) for 1 h to synthesize hydrogel.

### Preparation of silica nanoparticles
The silica nanoparticles were prepared according to the previously reported Stöber method. A typical synthesis of 50 nm nanoparticles was detailed as follows. Ethanol (740 mL), $H_2O$ (72 mL), and ammonium hydroxide solution (26 wt% in water; 30 mL) were added into a round bottom flask and stirred gently for half an hour. After that, TEOS (60 mL) was rapidly added to the flask, and the solution was kept at 30 °C for 6 h under magnetic stirring to finish the reaction. The

nanoparticle products were dispersed in ethanol after centrifuging and washing several times with ethanol. The silica nanoparticles with other diameters were synthesized by modifying the amount of $H_2O$ and TEOS. The solution was kept at 30 °C for 6 h under magnetic stirring. The nanoparticles in the whole solution were collected by centrifugation and the residual reactants were removed by ethanol washing.

## Preparation of ANP

Briefly, silica nanoparticles (1 g) were well dispersed in ethanol (1 L) by ultrasonic treatment and stirred gently for half an hour, then a DMF solution (10 mL) contained APTES (1 M), succinic anhydride (1 M) and DMAP (0.01 M) was added into nanoparticles dispersion and kept at 60 °C for under magnetic stirring for 12 h. The nanoparticles in the whole solution were collected by centrifugation and washed several times with ethanol to remove the residual reactants.

## Adhesion energy measurement

All tests were conducted in ambient air at room temperature. The effect of dehydration is not significant in the several minutes of the test. The adhesion energy between the hydrogel and rigid substrates was measured according to a 90-degree peeling test (ASTM D3330). The test was performed in a mechanical testing machine (Instron 5565A) with 500 N load cells and a 90° Angle Peel Fixture (Instron 2820-035). All rigid substrates were prepared with dimensions 15 cm in length, 5 cm in width and 2 cm in thickness. Hydrogels were cut to a size of $100 \times 25 \times 2$ mm³, and PC films were fixed to hydrogels as stiff backing to limit deformation to the crack tip thus all the work done by the machine would be equal to the energy dissipated at the crack tip. The hydrogel samples adhered to a substrate with one end open, and the free ends were fixated to the machine grip. The adhesion energy was the plateau value of the ratio of the force and width. The test of adhesion energy on PTFE was carried out by attaching nanohesives seon PTFE, attaching Scotch on PTFE, and attaching dissipative hydrogel on PTFE by cyanoacrylates.

The adhesion energy between the hydrogel and soft substrates was measured with 180-degree peeling tests (ASTM F2256). Hydrogels were cut to a size of $100 \times 25 \times 2$ mm³, and PC films were fixed to hydrogels as stiff backing. The hydrogel sample and the substrate adhered to a substrate with one end open, and the free ends were fixated to the machine grip. The adhesion energy was two times the plateau value of the ratio of the force and width.

## Measurement of fracture toughness of hydrogel

The matrix toughness was measured with pure shear tests. In brief, rectangular specimens ($50 \times 5 \times 2$ mm³) were fixed and stretched by an Instron machine (5565A with a load cell of 500 N). For notched samples, an edge crack of length 20 mm was cut using a razor blade in the middle of the gauge section of the specimen. The stretch rate was fixed at 2 min⁻¹. From the stress-stretch curves of the unnotched and notched specimens, the fracture energy was calculated following the method reported previously.

## Cell compatibility study

The nanohesives were incubated in a cell culture medium at 37 °C for 24 h to prepare a conditioned medium. Human gingival fibroblasts were plated in 96-well plates ($2.0 \times 10^4$ cells per well, $n = 3$ per group) and cultivated with each conditioned media (common culture medium and nanohesives conditioned medium 200 μL per well) for 24 h and 48 h. Cell viability was determined by a CCK-8 assay, and thus CCK-8 reagent was mixed with DMEM at 1:10 and added to wells (110 μL per well). Then the cells were cultivated for another 2 h at 37 °C. Finally, the culture media were removed and the viability was measured by a microplate reader at a wavelength of 450 nm.

## Anti-bacteria performance test

S. aureus (ATCC 25923) was employed to evaluate the antimicrobial activity of the nanohesives. The antibacterial properties were determined by optical density (OD) value. All the microbial cell suspensions were diluted to $10^5$ CFU/mL. The materials were punched into uniform discs of 6 mm diameter and then were sterilized using a high-pressure steam sterilizer (HVE-50, HIRAYAMA Company, Japan). The specimens were placed into the bottom of a 96-well plate, and 150 μL bacteria suspensions were added to each well and cultured for 24 h ($n = 3$ per group). Finally, the specimens were removed, and the absorbance at 600 nm was measured with an automatic microplate reader (MQX200, Bio-Red Company, America).

## Fabrication of the strain sensor

The structure of the strain sensor is sandwich-like, which is the bottom PI layer/metal layer/top PI layer. The fabrication process of the strain sensor starts with the preparation of the bottom PI layer (4–8 μm). Before spinning coating (4500 r/min) and baking the PI layer, a thin layer of PMMA (polymethylmethacrylate) is spinning coated (3000 r/min) on the silicon wafer acting as the adhesive layer to reinforce the interface strength between the silicon wafer and the PI layer. Then, Cr (10 nm) and Au (150 nm) are deposited sequentially on the PI covered silicon wafer by E-beam, in which Cr is acting as the transition layer and Au is the functional layer. After the Au and Cr are patterned by photolithography and wet etching sequentially, the residual photo resistor is removed and another layer of PI, i.e., the top PI layer, is spinning coated (4500 r/min, 4–8 μm) on the Au/Cr layer. The device is then transfer printed onto a PDMS film made temporary substrate, and both the top and bottom PI layers are patterned by RIE ($O_2$, 100 W for 2 h) using a mask of photolithography-patterned Cu layer (100 nm). In the end, removing the Cu mask by wet etching finishes the fabrication process.

## In vitro pulse simulation set-up

The pulse simulation set-up consists of a one-end sealed silicon rubber artificial artery (diameter: 4 mm) in combination with a force gauge that linked to the Instron machine (5565A with a load cell of 10 N); the artificial artery was assembled to a syringe pump to provide regularly reciprocating pressure. The strain sensor adhered beforehand on nanohesives was tightly wrapped around and fixed on the artificial artery, and characterized by changing the frequency and amplitude of the supplied liquid to the tube. The Instron machine recorded the pulsations of the pump and generated force on the sensor. Measurements were performed at a controlled temperature ($23 \pm 1$ °C) and atmospheric humidity ($50 \pm 10\%$ relative humidity).

## In vivo pulse monitoring

In vivo pulse monitoring of the femoral artery was performed in an adult Labrador dog. The dog was first anesthetized with ketamine hydrochloride (3 mg per kg body weight intramuscularly) and then with 0.3% pentobarbital sodium (1 mL per kg body weight) by intravenous injection to maintain general anesthesia. All materials used for implantation were sterilized in advance. The strain sensor was picked up by adhering to a 10 mm wide nanohesives patch. After brushing the ANP dispersion on the surface of the vessel, the strain sensor was wrapped around and fixed on the femoral vessel.

## Sample characterization

Zeta potential and size distribution were collected on Malvern Nano-ZS90. SEM images were measured using a field emission scanning electron micro analyzer (Zeiss Merlin Compact) at an acceleration voltage of 5 kV. The contact angles were tested by a Dataphysics OCA-25 contact angle analyzer. The electrical resistance and mechanical stability of the devices were tested by a

Keithley DMM7510 1/2 digit multimeter and mechanical system (Instron 5565A).

**Reporting summary**

Further information on research design is available in the Nature Portfolio Reporting Summary linked to this article.

## Data availability

All data are available in the main text or the Supplementary Information. All other data are available from the corresponding author upon request.

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

## Acknowledgements

This work was supported by the National Natural Science Foundation of China(Grants 22293044, U1932213, U20A6001, and 12172346), the National Key Research and Development Program of China (Grants 2018YFE0202201 and 2021YFA0715700), the University Synergy Innovation Program of Anhui Province (Grant GXXT-2019-028), Science and Technology Major Project of Anhui Province (201903a05020003), the China Postdoctoral Science Foundation (2021M703067) and the Zhejiang Provincial Natural Science Foundation of China (Grant No. LQ20B010006). This work has been supported by New Cornerstone Science Foundation.

## Author contributions

Z.P. and S.H.Y. conceived the idea and designed the experiments, Q.Q.F. and M.H.W. designed and carried out the experiments of blood flow monitoring experiments. S.H.Y. supervised the research. H.L.G., L.D., and Y.C., J.C.H., D.H.Z. helped characterizations and provided valuable advice. Z.P., P.Z., and D.D.C. performed the adhesion performance experiments and analyzed the data. Z.P., Q.Q.F. X.F., and S.H.Y. co-wrote the manuscript. All authors discussed the results.

## Competing interests

The authors declare no competing interests.
