## [Peer Review File · Nature Communications]

Designing nanohesives for rapid, universal and robust hydrogel adhesionREVIEWER COMMENTS

Reviewer #1 (Remarks to the Author):

In this work, the authors develop nanohesives through surface-activated nanoparticles (ANP) and dissipative hydrogel. Starting from the actual adhesion effect, nanohesives show excellent broad-spectrum adhesion performance. However, there are many points to be addressed before it is in an acceptable form. I recommend its reconsideration if the following issues are properly responded.

1. The author mentioned that the scattered hydrogel designed can quickly absorb the water at the interface, thereby achieving the interconnection adhesion of ANP and the adhered in the second. The author mentioned that the scattered hydrogel designed can quickly absorb the water at the interface, thereby achieving the interconnection adhesion of ANP and the adhered.
2. In the adhesion of hydrogel, the dispersing matrix of the hydrogel itself can provide irreplaceable effects for adhesion. The author mentioned related expressions, but lack of specific data to support the increase of fracture toughness on hydrogel. The author should compare the advantages of the related water gel broken toughness and adhesion to explain the advantages of nanohesives.
3. The author mentioned that nanohesives can achieve the establishment of adhesion within seconds. From the application of glue (ANP) to the establishment of adhesion, what is the specific impact of time? What is the trend?
4. In the hydrogel adhesion of nanohesives, the author mentioned that the surface of the nanoparticles on the interface is mainly in two states: bridging state and Pickering state. On the surface of engineering materials (metal, ceramic, plastic, and rubber) and biological tissues, which state plays a leading role?
5. We all know that direct monitoring of medical equipment for patients needs to be demolished after the patient recovery. Do nanohesives have the possibility and recycling of disassembly?

Reviewer #2 (Remarks to the Author):

Nanoparticles with carboxyl groups are used as glue between dissipative hydrogels and various substrates. Authors claimed that adhesion energy was significantly improved compared with nanoparticle adhesives in previous reports due to the hydrogen bonding, van der Waals interaction, and dissipative energy. However, the significant contribution to such a huge improvement may not be from the nanoparticles but from the dissipative hydrogel. The contribution of dissipation energy to adhesion energy has been known.

Authors should measure the adhesion energy of nanoparticles between non-dissipative hydrogels such as PDMA and substrates for evaluating the contribution of the nanoparticles to adhesion energy. Then,

adhesion energies should be compared with values in previous reports.

If there is no significant contribution of the carboxyl groups on nanoparticles in such a huge improvement or synergistic effect with dissipative hydrogels, the manuscript should be published in a specialized journal or application-focused one rather than nature communication.

Reviewer #3 (Remarks to the Author):

In this manuscript, the authors developed a type of nanohesives that incorporates a chemically active nanoparticles and dissipative hydrogel to achieve a practical nanoparticle-based adhesive. The nanohesives can implement robust adhesion to different substrate surfaces, including metals, ceramics, plastics, rubbers and various biological tissues. In general, this is an interesting and meaningful study. But there some issues should be addressed before publication.

1. The authors should give more detailed explanation and direct data about the adhesion mechanism of the nanohesives. The author mentioned “absorption of the interfacial water by dissipative hydrogel densified the ANP and facilitate the formation of nanohesion”, what will happen if ANP glue with higher nanoparticle concentration was used?
2. How about the mechanical strength of the hydrogels in the nanohesives system? Are the hydrogels degradable?
3. What happened when the nanohesives was peeled from the substrates? Were the nanoparticles detached from the substrates or the hydrogels broken?
4. The quantitative analysis of the size of nanoparticles should be performed.
5. Many pictures or the text in the figures are unclear, e.g., Figure 2a.

We are truly grateful to the insightful comments and thoughtful suggestions. Based on these valuable comments and suggestions, we have diligently addressed each one and made careful modifications on the original manuscript. All changes made to the text are in red color. We hope the new manuscript will meet the journal's standard. Below, we provide our point-by-point responses to the comments and questions raised by the reviewers:

REVIEWER COMMENTS

Reviewer #1 (Remarks to the Author):

In this work, the authors develop nanoheives through surface-activated nanoparticles (ANP) and dissipative hydrogel. Starting from the actual adhesion effect, nanoheives show excellent broad-spectrum adhesion performance. However, there are many points to be addressed before it is in an acceptable form. I recommend its reconsideration if the following issues are properly responded.

Our Response: We thank the reviewer for the positive evaluation and the constructive comments on the manuscript. We address the comments below.

Reviewer's Comment 1: The author mentioned that the scattered hydrogel designed can quickly absorb the water at the interface, thereby achieving the interconnection adhesion of ANP and the adhered in the second. The author mentioned that the scattered hydrogel designed can quickly absorb the water at the interface, thereby achieving the interconnection adhesion of ANP and the adhered.

Our Response: We appreciate the reviewer's comment.

In this study, we present the design of a novel class of nanoheives that utilize the modulation of hydrogel mechanics and surface chemical activation of nanoparticles. These nanoheives enable the rapid formation of strong hydrogel adhesion within seconds, onto various engineering solids and biological tissues, without the need for any surface pre-treatments.

The development of these nanohesives represents a promising strategy for addressing the challenges associated with hydrogel adhesion. Their ability to achieve rapid and robust adhesion to diverse surfaces, combined with their biocompatibility and antimicrobial properties, makes them highly valuable for various applications in the field of hydrogel-based engineering.

Reviewer's Comment 2: In the adhesion of hydrogel, the dispersing matrix of the hydrogel itself can provide irreplaceable effects for adhesion. The author mentioned related expressions, but lack of specific data to support the increase of fracture toughness on hydrogel. The author should compare the advantages of the related water gel broken toughness and adhesion to explain the advantages of nanohesives.

Our Response: Thank you for your suggestion.

The advantages of the fracture toughness of hydrogel on the adhesion energy were carried out in this response and relevant data was added in the revised manuscript. In the nanohesives developed in this article, one of the most dominative factor towards the interfacial adhesion energy of hydrogel bonding is the dissipative capability of hydrogel. To verify the influence of the dissipative capability of hydrogel, we prepared a series of hydrogel with same components but various contents of agarose, which further formed the dissipative network in the hybrid hydrogel. Through pure-shear test, we characterized the fracture energy of the hydrogels with different contents of agarose, ranging from 0, 0.5 wt%, 1.0 wt%, 1.5 wt%, to 2.0 wt%. The results indicated that the increase of agarose contents, substantially enhance the fracture toughness of hydrogel, which represents the dissipative capability of hydrogel (**Fig. R1a**). The results indicate that the dissipative capacity of the hydrogel increased, the adhesion energy of the nanohesives was correspondingly enhanced.

Furthermore, in addition to the dissipative network, the presence of MOTAC (cationic component) in the long-chain network can enhance the hydrophilicity of the chains, leading to a decrease in the mechanical performance of the hydrogel. By regulating the content of MOTAC, we observed a decrease in the overall fracture toughness of the dissipative hydrogel (**Fig. R1b**). However, simultaneously, the

increase in cationic groups enhances the interaction density between the dissipative hydrogel and the nanoparticles, thereby improving the adhesive performance. The antagonistic effects of decreased fracture toughness and increased bonding sites result in a non-monotonic variation in adhesive energy with increasing MOTAC content, as illustrated in **Fig. R1b**. These results from both tests strongly support the reviewer's suggestion that regulating fracture toughness is beneficial for optimizing the adhesive performance of nanohesives.

We have added the discussion on the fracture energy of dissipative hydrogels in the main text on Page 8, Lines 11-19, and a figure in the supplementary information as Supplementary Fig. 4.

“Furthermore, the interfacial adhesion energy of hydrogel bonding in nanohesives is predominantly influenced by the dissipative capability of the hydrogel. To investigate this influence, hydrogels with varying agarose contents, which forming a dissipative network, were prepared. The fracture toughness of the hydrogels increased with increasing agarose contents, indicating enhanced dissipative capability (Fig. 2j). Additionally, the presence of MOTAC in the hydrogel network improved adhesion but decreased mechanical performance. The overall adhesive energy exhibited a non-monotonic variation with increasing MOTAC content, suggesting the tradeoff between fracture toughness and bonding sites (Supplementary Fig. 4).”

Fig. R1. The influence of component modulation on the fracture toughness of the dissipative hydrogel and the adhesive energy of nanohesives. a, Both fracture toughness and adhesive energy increase with an increase in the concentration of agarose. **b,** The fracture toughness decreases with an increase in MOTAC content, while the adhesive energy exhibits a non-monotonic variation. Values in panel represent the mean and the standard deviation ($n=3-4$).

Reviewer's Comment 3: The author mentioned that nanohesives can achieve the establishment of adhesion within seconds. From the application of glue (ANP) to the establishment of adhesion, what is the specific impact of time? What is the trend?

Our Response: Thank you for raising these questions.

In the nanohesives developed in this study, the establishment of stable adhesion occurs within a short time frame due to the rapid water absorption property of the dissipative hydrogel at the interface. Interestingly, the instantaneous adhesion energy is comparable to the adhesion energy achieved over a longer period of time, highlighting the efficient and effective nature of the adhesive process.

We propose that the mechanism underlying the formation of nanohesion is an absorption-interaction process. For adhesion of hydrogels and other surface-wetted substrates, such as biological tissues, a common strategy is the diffusion-based mechanism (*J. Mech. Phys. Solids*, 2020, 137, 103863; *Science*, 2017, 357, 378; *Adv. Mater.* 2018, 30, 1800671.). In this mechanism, adhesive components, such as monomers, macromolecules, or polymers, diffuse through the interfacial water layer from the adhesive to form physical/chemical interactions with the substrate surface. This diffusion-based mechanism typically requires a relatively long time due to the slow diffusion rate of large molecules. For instance, tough hydrogel adhesives and topological adhesives often require 5-30 minutes to form stable bonding (*Science*, 2017, 357, 378; *Adv. Mater.* 2018, 30, 1800671.).

In contrast, in this study, the introduction of charged functional groups into the hydrogel matrix significantly enhances its water-absorbing rate (*J. Chem. Phys.* 1990, 93, 4462; *J. Chem. Phys.* 2005, 122, 154903). This is primarily due to increased hydrophilicity, which promotes water absorption by creating stronger affinity between the hydrogel and water molecules. The electrostatic interactions between the charged functional groups and water molecules further contribute to swelling in the hydrogel, enabling it to absorb water more rapidly. The charged functional groups also induce osmotic pressure, driving the influx of water into the hydrogel. Furthermore, the ANP solution used in our experiments exhibits relatively low water content (less than 80 wt.%). The regulation of water absorption in the nanohesives system can be influenced

by various factors such as monomer type, concentration variations, and changes in water content. In our future work, we plan to investigate these factors in detail to gain a better understanding of their impact on water absorption properties in nanohesives. Distinct from the diffusion process observed in traditional adhesives, the dissipative hydrogel rapidly absorbs the interfacial water layers from both the glue and the substrate surface. Consequently, the density of the dispersed nanoparticle aggregates in the ANP glue increases due to dehydration, facilitating their intimate contact with the substrate. The interactions between the nanoparticles and the substrate primarily involve physical interactions (*Intermolecular and Surface Forces, Academic Press, Amsterdam, Elsevier, 2011*), enabling the rapid establishment of stable adhesion within a notably short time frame.

Therefore, the specific impact of time on nanohesive bonding can be observed in the adhesive energy-contact time tests. The trend indicates that the adhesion energy of nanohesives remains relatively constant across different contact times, including immediate bonding (within 2 seconds) and longer contact time intervals (**Fig. R2**). This implies that the adhesion reaches its maximum toughness within a very short period, and further prolonging the contact time has a slight impact on the adhesive strength. In other words, the rapid formation of nanohesion allows for tough bonding in seconds, eliminating the need for extended contact times typically associated with traditional adhesive systems.

We have added the discussion on the absorption-interaction adhesion formation process in the main text on Page 9, Lines 4-17, and a figure in the supplementary information as Supplementary Fig. 5a.

“We propose that the mechanism underlying the formation of nanohesion is an absorption-interaction process. In this study, the introduction of charged functional groups into the hydrogel matrix significantly enhances its water-absorbing rate. The dissipative hydrogel rapidly absorbs the interfacial water layers from both the adhesive and the substrate surface. Consequently, the density of the dispersed nanoparticle aggregates in the ANP glue increases due to dehydration, facilitating their intimate contact with the substrate. The absorption-interaction mechanism underlying

nanoheesion formation facilitates extremely rapid bonding, as evidenced by our experimental tests (Supplementary Fig. 5a).”

Fig. R2 The relationship between adhesion time and adhesion energy. Values in panel represent the mean and the standard deviation ($n=3-4$).

Reviewer’s Comment 4: In the hydrogel adhesion of nanoheesives, the author mentioned that the surface of the nanoparticles on the interface is mainly in two states: bridging state and Pickering state. On the surface of engineering materials (metal, ceramic, plastic, and rubber) and biological tissues, which state plays a leading role?

Our Response: We sincerely appreciate the insightful comments provided by the reviewer.

Based on previous literature reports, we believe that the dominance of a particular state (bridging or Pickering) is likely influenced by the elastic modulus of both the nanoparticles and the substrate. In this study, based on the SEM observations of the interface between dissipative hydrogel-ANP glue and the skin tissue, we have described two distinct states of nanoparticles, namely, the bridging state and the Pickering state. Previous studies have investigated the distribution of nanoparticles at the interface using molecular simulations and theoretical calculations, providing valuable insights into the various states of nanoparticles in contact with substrates (*Langmuir*, 2019, 35, 7277; *Macromolecules*, 2016, 49, 3586). It has been established that nanoparticles in contact with elastic substrates can adopt two distinct states: the Pickering state, in which

they are distributed between two contacting gels, and the bridging state, in which they form a connection between the two substrates, spanning a gap.

Furthermore, simulation analysis has provided evidence supporting the preference of the Pickering state for smaller and softer nanoparticles that interact with soft substrates, whereas the bridging state is favored in the case of larger and more rigid nanoparticles interacting with hard substrates. (*Langmuir*, 2019, 35, 7277; *Macromolecules*, 2016, 49, 3586). In our study, as we employed silica nanoparticles, which are relatively rigid inorganic nanoparticles, with higher elastic modulus compared to the dissipative hydrogel, biological tissues, and soft materials, we believe that the Pickering state is more favorable when encountering soft substrates. On the other hand, for hard materials such as metals and ceramics, the bridging state is more likely to prevail.

Comment 5: We all know that direct monitoring of medical equipment for patients needs to be demolished after the patient recovery. Do nanohesives have the possibility and recycling of disassembly?

Our Response 5: Thank you for the excellent suggestion.

It is indeed necessary to address the removal of non-absorbable monitoring devices in a gentle manner. In this regard, we have investigated the application of a triggering solution, specifically a 0.5 M aqueous solution of Sodium Bicarbonate (SBC), for facilitating the gentle detachment of our nanohesives in this response (Fig. R3a).

This triggerable detachment mechanism is based on the cleavage of the bonding interactions between the nanohesives and the tissue surface. The results demonstrate that the adhesive energy of the nanohesives significantly decreases when the tight non-covalent bonds between the nanohesives and substrates are disrupted by the introduction of SBC. Particularly, the conversion of the carboxyl groups on ANP to sodium carboxylate contributes to the observed reduction in adhesive energy of the nanohesives, as illustrated in **Fig. R3b**.

To enable the reuse of nanohesives, we performed a thorough rinsing of the detached surfaces previously bonded with ANP-containing dissipative hydrogel, thereby removing any residual SBC on the surface. Subsequently, the cleaned nanohesives were

re-applied onto the substrate surface, and it was observed that the reused nanohesives retained their adhesive energy over 500 J/m^2 . This demonstrates the capability of nanohesives to maintain their adhesive strength even after undergoing the detachment and reapplication process.

In this response, we have demonstrated the tunable delamination and recyclability of nanohesives using glass as a representative substrate. Due to the broad spectrum nature of nanohesives, it is anticipated that these properties can be extended to applications involving various other adhesive substrates. In future studies, we will focus on investigating these properties in the context of bio-electronic devices specifically designed for specific biological tissues and organs.

Fig. R3 a. Photographs for instant robust adhesion and triggerable benign detachment of the nanohesives in vitro. **b,** The adhesion energy of original formed nanohesion, SBC triggered detached nanohesion, and recycling used nanohesion, respectively. Values in panel represent the mean and the standard deviation ($n=3-4$).

Reviewer #2 (Remarks to the Author):

Nanoparticles with carboxyl groups are used as glue between dissipative hydrogels and various substrates. Authors claimed that adhesion energy was significantly improved compared with nanoparticle adhesives in previous reports due to the hydrogen bonding, van der Waals interaction, and dissipative energy. **However, the significant contribution to such a huge improvement may not be from the nanoparticles but from the dissipative hydrogel.** The contribution of dissipation energy to adhesion energy has been known.

Authors should measure the adhesion energy of nanoparticles between non-dissipative hydrogels such as PDMA and substrates for evaluating the contribution of the nanoparticles to adhesion energy. Then, adhesion energies should be compared with values in previous reports.

If there is no significant contribution of the carboxyl groups on nanoparticles in such a huge improvement or synergistic effect with dissipative hydrogels, the manuscript should be published in a specialized journal or application-focused one rather than nature communication.

Our Response: We are thankful for the reviewer's time and the constructive comments on the manuscript. We address the comments below.

1. Interfacial Interactions and Mechanical Dissipation.

The contribution of the dissipation capability of hydrogel to adhesion energy has been well-known in previous report (*Nat. Mater.* 2016, 15, 190.). However, by disregarding the effects of mechanical dissipation within the solid and friction at the interface, the overall interfacial toughness of the bonding between the hydrogel and solid can be represented as follows (*Nat. Mater.* 2016, 15, 190.).

$$\Gamma = \Gamma_0 + \Gamma_D$$

In above equation, while the intrinsic adhesion toughness Γ_0 may be significantly lower than the dissipative adhesion toughness Γ_D in hydrogel interfacial bonding, it remains essential to provide cohesion at hydrogel-solid interface, while enabling extensive deformation and mechanical dissipation within the bulk hydrogel, leading to elevated values of Γ_D .

In the nanohesives, the intrinsic adhesive bonding is provided by the interfacial interactions brought by activated nanoparticles (ANP), which enables the deformation and dissipation within the dissipative hydrogels. To demonstrate the strong interfacial interaction, we employed ANP glue to reconnect a fractured dissipative hydrogel (**Fig. R4**). The results reveal that the hydrogel reconnected with ANP exhibited tensile properties comparable to those of an intact hydrogel. Furthermore, the re-fracture site did not occur at the original fracture location. This outcome highlights the remarkable strength of the interfacial interaction provided by the nanoparticles, forming the foundation to drive the hydrogel's dissipative mechanism.

Fig. R4 The ANP adhesive is utilized for the reconnection of dissipative hydrogels.

2. Comparison of adhesion performance on PDMA hydrogel.

To further demonstrate the distinction of adhesive performance between carboxyl-functionalized nanoparticles glue used in this study and previously reported nanoparticle-based adhesives, we compared two examinations on a widely used PDMA (Poly *N,N*-dimethylacrylamide) hydrogel (**Table R1**). As shown in **Fig. R5**, when ANP adhesive was utilized, the strength achieved in lap-shear tests reached 23 kPa, while in the 90-degree peel tests, it reached 400 J/m². These results significantly outperform the adhesive performance reported in previous literature (**Table R2**).

Table R1. The formulation for synthesizing PDMA hydrogel.

DMA	MBA 2 wt% aqueous solution	I -1173	DIW
3.465 g	81 μ L	57 mg	10 mL

Table R2. The adhesive performance of ANP glue in adhering PDMA hydrogels is compared with previous reported nanoparticles-based glues.

Literatures	Nanoparticles	Lap-shear strength	Adhesion energy
Nature, 2014, 505, 382	Silica Nanoparticles		6-10 J/m ²
Acta Biomater., 2017, 57, 404	HAp nanoparticles	2-8 kPa	
Acta Biomater., 2022, 152,171	PDA-Mesoporous silica nanoparticles-PVA	1 kPa	
Chem. Mater., 2022, 34, 584	Colloidal supraballs of mesoporous silica nanoparticles		10-100 J/m ²
ACS Appl. Mater. Interfaces, 2017, 9, 31469	Colloidal mesoporous silica nanoparticles		5-35 J/m ²
This article	Activated silica nanoparticles	~23 kPa	~400 J/m²

Fig. R5 a, The typical curve in the 90 degree peeling test of PDMA adhered by ANP glue. **b,** Photograph of specimens under peeling test. **c,** The typical curves in lap-shear test of PDMA adhered by ANP glue. **d,** Photographs of specimens under lap-shear test.

3. Influence of carboxyl functionalization on the adhesive performance.

In the main text, Figure 2f illustrates the differences in adhesive energy resulting from the interactions between various nanoparticle-hydrogel combinations. When comparing the combinations that lack of carboxyl groups, which contribute to electrostatic interactions, both the group of carboxyl nanoparticle and non-charged polymer chain hydrogel ($13.06\pm 1.49 \text{ J/m}^2$), as well as the group of hydroxyl nanoparticle and charged hydrogel ($229.62\pm 33.12 \text{ J/m}^2$), exhibited significantly lower adhesive energies compared to the group of ANP and charged dissipative hydrogel system claimed in the paper ($\sim 1300 \text{ J/m}^2$).

In this response, we specifically focus on comparing the influence of different nanoparticle surface chemical properties on adhesive energy. As depicted in the **Fig. R7**, the surface carboxylation treatment significantly enhances the interfacial adhesion energy in comparison to the original hydroxylated surface ($229.62\pm 33.12 \text{ J/m}^2$) and the surface modified with amino groups ($188.24\pm 8.99 \text{ J/m}^2$). Consistent with the earlier discussion, we attribute this enhancement to the improvement in intrinsic interfacial toughness, which amplifies the dissipative capability of the hydrogel phase, ultimately leading to an overall increase in adhesive energy. H_2

We would like to express our gratitude to the reviewer for their valuable suggestions in helping us refine the discussion on the enhancement of intrinsic interfacial energy through nanoparticle surface chemical modifications, which provides synergistic effect with dissipative hydrogels to promote interfacial adhesion.

Fig. R6 Different surface chemical properties of nanoparticles result in varied adhesive performance.

Reviewer #3 (Remarks to the Author):

In this manuscript, the authors developed a type of nanohesives that incorporates a chemically active nanoparticles and dissipative hydrogel to achieve a practical nanoparticle-based adhesive. The nanohesives can implement robust adhesion to different substrate surfaces, including metals, ceramics, plastics, rubbers and various biological tissues. In general, this is an interesting and meaningful study. But there some issues should be addressed before publication.

Our Response: We thank the reviewer for the positive evaluation and the constructive comments on the manuscript. We address the comments below.

Reviewer's Comments 1: The authors should give more detailed explanation and direct data about the adhesion mechanism of the nanohesives. The author mentioned “absorption of the interfacial water by dissipative hydrogel densified the ANP and facilitate the formation of nanohesion”, what will happen if ANP glue with higher nanoparticle concentration was used?

Our Response: We appreciate the insightful comments from the review. We apologize for any inaccuracies in conveying our adhesive mechanism.

The interfacial toughness in hydrogel adhesion can be quantified using the following equation (*Nat. Mater.*, 2016, 15, 190.).

$$\Gamma = \Gamma_0 + \Gamma_D$$

Here, Γ_0 denotes the intrinsic adhesive toughness, while Γ_D represents the dissipative toughness attributed to the overall deformation of the hydrogel. Previous studies on nanoparticle-based adhesives have highlighted the crucial role of affinity adsorption between polymer chains within the hydrogel and the nanoparticle surface as the basis for nanoparticle adhesion (*Nature*, 2014, 505, 382; *Soft Matter*, 2019, 15, 9942; *Macromolecules*, 2021, 54, 1992.). In this study, we capitalized on this fundamental principle by implementing chemical modifications to the polymer chains in the hydrogel, introducing positively charged groups into the molecular structure. In

contrast to prior investigations that employed nanoparticles with less active surface groups, such as hydroxyl groups (*Nature*, 2014, 505, 382; *Acta Biomater.*, 2017, 57, 404; *Chem. Mater.*, 2022, 34, 584.), we performed chemical modifications on the nanoparticle surface to generate activated nanoparticles capable of robust interactions with the positively charged groups within the hydrogel. Through these dual modifications, the bonding at the hydrogel adhesion interface is significantly enhanced (**Fig. 2f**). Furthermore, while previous reports predominantly focused on nanoparticle adhesion within single-network hydrogels like PDMA hydrogels, our study incorporates a toughening mechanism by incorporating a dissipative polymer network into the long-chain polymer network of the hydrogel. This integration greatly amplifies the energy dissipation capability of the hydrogel during deformation, consequently elevating the dissipative toughness of the hydrogel adhesion interface (**Fig. 2i**). By simultaneously enhancing the intrinsic adhesive toughness represented by Γ_0 and the dissipative toughness denoted by Γ_D , we ultimately achieved a synergistic effect that enhances the toughening of the interfacial adhesion.

We propose that the mechanism underlying the formation of nanohesion is an absorption-interaction process. For adhesion of hydrogels and other surface-wetted substrates, such as biological tissues, a common strategy is the diffusion-based mechanism. In this mechanism, adhesive components, such as monomers, macromolecules, or polymers, diffuse through the interfacial water layer from the adhesive to form physical/chemical interactions with the substrate surface. This diffusion-based mechanism typically requires a relatively long time due to the slow diffusion rate of larger molecules. For instance, tough hydrogel adhesives and topological adhesives often require 5-30 minutes to form stable bonding (*Science*, 2017, 357, 378; *Adv. Mater.* 2018, 30, 1800671.).

In contrast, in this study, the introduction of charged functional groups into the hydrogel matrix significantly enhances its water-absorbing rate (*J. Chem. Phys.* 1990, 93, 4462; *J. Chem. Phys.* 2005, 122, 154903). This is primarily due to increased hydrophilicity, which promotes water absorption by creating stronger affinity between the hydrogel and water molecules. The electrostatic interactions between the charged

functional groups and water molecules further contribute to the swelling in the hydrogel, enabling it to absorb water more rapidly. The charged functional groups also induce osmotic pressure, driving the influx of water into the hydrogel. Furthermore, the ANP solution used in our experiments exhibits relatively low water content (less than 80 wt.%). Distinct from the diffusion process observed in traditional adhesives, the dissipative hydrogel rapidly absorbs the interfacial water layers from both the adhesive and the substrate surface. Consequently, the density of the dispersed nanoparticle aggregates in the ANP glue increases due to dehydration, facilitating their intimate contact with the substrate. The ensuing interactions between the nanoparticles and the substrate primarily involve physical interactions (*Intermolecular and Surface Forces*, Academic Press, Amsterdam, Elsevier, 2011), enabling the rapid establishment of stable adhesion within a notably short time frame.

The absorption-interaction mechanism underlying nanohesion formation facilitates extremely rapid bonding, as evidenced by our experimental tests (**Fig. R7a**). The adhesive strength of nanohesion showed no significant difference between immediate bonding group (within 2 seconds) and longer contact time groups. This observation indicates that nanohesives can indeed be characterized as instant adhesives. In the adhesive energy-contact time tests, we utilized a 20 wt% dispersion of ANP. While higher concentrations may potentially result in faster bonding speeds, the practical significance of such acceleration might be challenging to discern. For the higher concentration groups, a slight decrease in adhesion energy was observed. This could be attributed to an excessive accumulation of nanoparticles at the adhesive interface due to the higher concentration. Conversely, when using lower concentrations of the dispersion, the adhesive energy tends to decrease (**Fig. R7b**). We attribute this effect to a reduction in the distribution density of nanoparticles at the interface as the concentration decreases, thereby diminishing the number of contact points for interfacial adhesion and consequently reducing the intrinsic bonding toughness.

Fig. R7 a, The relationship between adhesive energy and adhering time. **b**, The relationship between adhesion energy and the concentration of ANP glue. Values in panel represent the mean and the standard deviation ($n=3-4$).

We would like to express our gratitude once again for the valuable insights provided by the reviewers regarding the mechanism of nanohesives. We hope this response could address the reviewers' suggestions by providing explanations on the enhancement of adhesion toughness and the mechanisms underlying nanohesion formation.

We have added the discussion on the absorption-interaction adhesion formation process in the main text on Page 9, Lines 4-17, and a figure in the supplementary information as Supplementary Fig. 5a.

“We propose that the mechanism underlying the formation of nanohesion is an absorption-interaction process. In this study, the introduction of charged functional groups into the hydrogel matrix significantly enhances its water-absorbing rate. The dissipative hydrogel rapidly absorbs the interfacial water layers from both the adhesive and the substrate surface. Consequently, the density of the dispersed nanoparticle aggregates in the ANP glue increases due to dehydration, facilitating their intimate contact with the substrate. The absorption-interaction mechanism underlying nanohesion formation facilitates extremely rapid bonding, as evidenced by our experimental tests (Supplementary Fig. 5a). While higher concentrations may potentially result in faster bonding speeds, the practical significance of such acceleration might be challenging to discern. Conversely, when using lower concentrations of the dispersion, the adhesive energy tends to decrease (Supplementary

Fig. 5b). We attribute this effect to a reduction in the distribution density of nanoparticles at the interface as the concentration decreases, thereby diminishing the number of contact points for interfacial adhesion and consequently reducing the intrinsic bonding toughness.”

Comments 2. How about the mechanical strength of the hydrogels in the nanohesives system? Are the hydrogels degradable?

Our Response: Many thanks for the excellent suggestion.

For dissipative hydrogels, mechanical performance is a crucial aspect in adhesive applications. In this study, we characterized the tensile behavior and fracture toughness of the hydrogel (2 wt.% agarose, 30 wt.% MOTAC). The tensile curve of the hydrogel exhibited elastic and plastic deformation regions (**Fig. R8a**), with a maximum tensile stress of approximately 63.33 ± 3.75 kPa and an elastic modulus of around 67.61 ± 3.56 kPa (**Fig. R8b**). This performance is a result of the combined effects of the long-chain network and the dissipative network, which also remain valid for fracture toughness. Increasing the agarose content in the hydrogel led to an increase in the dissipative network, resulting in enhanced fracture toughness (**Fig. R8c**). Consequently, the overall adhesive energy of the nanohesives increased. On the other hand, the presence of MOTAC in the long-chain network increased the hydrophilicity of the molecular chains, leading to a decrease in mechanical performance of the hydrogel. Therefore, increasing the MOTAC content resulted in a decrease in fracture toughness. However, due to the increased number of adhesive sites in the hydrogel as a result of higher MOTAC content, the effect of MOTAC on adhesive energy exhibited a non-monotonic trend (**Fig. R8d**).

Fig. R8 Mechanical performance of dissipative hydrogel. **a**, The typical curves of dissipative hydrogel in tensile test. **b**, The maximum strength and elastic modulus of dissipative hydrogel. **c**, both fracture toughness and adhesive energy increase with an increase in the concentration of agarose. **d**, the fracture toughness decreases with an increase in MOTAC content, while the adhesive energy exhibits a non-monotonic variation. Values in panel represent the mean and the standard deviation ($n=3-4$).

The degradation performance of hydrogels is of significant importance for their in vivo applications. However, in this study, the dissipative hydrogel was unable to degrade due to the absence of degradable crosslinking points or other degradation mechanisms. Nevertheless, we greatly appreciate the insightful suggestion from the reviewer. In this response, we have introduced degradable crosslinking points into the hydrogel by replacing *N,N'*-Methylene-bis-acrylamide with *N,N'*-Bis(acryloyl)cystamine, and utilized biodegradable natural polymer gelatin as a substitute for agarose in the dissipative network (**Table R3**). The crosslink agent with disulfide bonds is sensitive to L-cysteine (5 mg/100 mL), while gelatin is sensitive to collagenase (5 mg/100 mL). This degradable hydrogel may meet the requirements of biodegradability in vivo. In vitro simulations were performed to characterize the two degradation mechanisms separately (**Fig. R9**). The results demonstrated that the hydrogel fulfilled the degradability requirements within the controlled time frame.

Table R3. The formulation for synthesizing degradable dissipative hydrogel.

DMA	BAC 2 w/v % aqueous solution	KGA	Gelatin 2 w/v% aqueous solution
3.46 g	130 μ L	50 mg	10 g

Fig. R9 In vitro biodegradation of dissipative hydrogel. The hydrogels were biodegraded in Dulbecco's phosphate-buffered saline (DPBS) with or without L-cysteine and collagenase. Values in panel represent the mean and the standard deviation ($n=3-4$).

Comments 3: What happened when the nanohesives was peeled from the substrates? Were the nanoparticles detached from the substrates or the hydrogels broken?

Our Response 3: Thank you for your valuable feedback on our manuscript.

When the dissipative hydrogel detached from the substrate surface, fracture occurred at the interface. Elemental analysis of the post-detachment hydrogel surface and substrate surface revealed a similar distribution of surface elements, indicating the presence of residual nanoparticles on the hydrogel surface and residual hydrogel polymer chains and nanoparticles on the substrate surface (**Fig. R10 a-b**). Furthermore, morphological characterization of the residue on the substrate surface using SEM demonstrates the presence of a thin layer of hydrogel-nanoparticle mixture residues along the direction of detachment (**Fig. R10 d-f**).

Fig. R10 Analysis of the detached surface. a and b, The digital photos of the detached nanohesives, and residues could be observed at both dissipative hydrogel side and the substrate side. **c,** Elementary analysis from X-ray photoelectron spectroscopy. The nitrogen was from the polymer chains in the hydrogels, and silicon was from the silica nanoparticles. **d,** Morphology analysis of the detached surface on substrates by scanning electron microscopy. Along the peeling direction, distinct ridges and basins can be observed. **e,** Both hydrogels and nanoparticles residues can be seen at the ridges. **f,** similar results were presented in the basins.

Comments 4: The quantitative analysis of the size of nanoparticles should be performed.

Our Response 4: Thank you for your valuable feedback on our manuscript.

We appreciate your time and effort in reviewing our work. We agree that performing a quantitative analysis of the size of nanoparticles would be beneficial to our study.

Therefore, in this response, we characterized the particle size information of the nanoparticles using two methods. Firstly, the morphology of the nanoparticles was characterized using TEM (transmission electronic microscope), as shown in **Figures R11 a to g**, revealing an increasing particle size trend. The corresponding hydration radii of these nanoparticles were determined through DLS (dynamic light scattering) measurements, and the statistical distribution is presented in **Fig. R11h**.

We have added the discussion on the absorption-interaction adhesion formation process in the main text on Page 7, Lines 14-15, and a figure in the supplementary information as Supplementary Fig. 3.

“The size characterization of nanoparticles was presented in Supplementary Fig. 3.”

Fig. R11 a-g, The TEM images of different sizes of ANPs. **h,** The distribution of hydrodynamic sizes of different ANPs measured by DLS.

Comments 5: Many pictures or the text in the figures are unclear, e.g., Figure 2a.

Our Response 5: Thank you for your suggestions on our manuscript.

We appreciate your time and effort in reviewing our work. We have carefully reviewed the figures in manuscript, especially Figure 2a, and have made revisions to improve the clarity of the pictures and text. We believe that the revised version of the figure is now clear and easy to read. We hope that these revisions have addressed your concerns, and we are grateful for your helpful feedback in improving the quality of our manuscript.

REVIEWERS' COMMENTS

Reviewer #1 (Remarks to the Author):

The authors have addressed my comments. I recommend publishing this manuscript on Nature Communications.

Reviewer #2 (Remarks to the Author):

Authors clarify the most of issues properly which I raised in first review. However, for fair comparison, I strongly believe that PDMA data should be included in the manuscript and discussed. In next revision, authors should add more information on PDMA such as swelling ratio and their young's modulus since that affect the adhesion energy as shown in reference 25. In addition, equation should be provided for calculating adhesion energy (J/m^2). After addressing these issues, manuscript may be accepted for publication.

Reviewer #3 (Remarks to the Author):

The authors have carefully revised the manuscript and addressed all the concerns. The manuscript has been improved a lot, and it is suitable for publication now.

REVIEWERS' COMMENTS

Reviewer #1 (Remarks to the Author):

The authors have addressed my comments. I recommend publishing this manuscript on Nature Communications.

Response: We greatly appreciate the valuable time and insightful comments from reviewer, which have significantly enhanced the quality of our work. Furthermore, we deeply appreciate the reviewer's recommendation, and we are honored to have the opportunity to publish our work in Nature Communications.

Reviewer #2 (Remarks to the Author):

Authors clarify the most of issues properly which I raised in first review. However, for fair comparison, **I strongly believe that PDMA data should be included in the manuscript and discussed.** In next revision, authors should add more information on PDMA such as **swelling ratio** and **their young's modulus** since that affect the adhesion energy as shown in **reference 25**. In addition, **equation should be provided for calculating adhesion energy (J/m^2).** After addressing these issues, manuscript may be accepted for publication.

Response: We greatly appreciate the valuable feedback and careful evaluation of our revised manuscript from the reviewer. We appreciate your suggestion regarding the inclusion of PDMA data and discussing their adhesion energy. We had made a revision in the manuscript in Page 12, Lines 4-7.

“Further comparison of ANP and previously reported nanoparticle-based glue, nanohesives presented in this study outperform the reported adhesive performance on widely used PDMA (Poly N,N-dimethylacrylamide) hydrogel (Supplementary Table

1).”

And we further added the content about synthesizing PDMA hydrogel in the Supplementary Methods.

“The PDMA hydrogel was synthesized by mixing DMA (3.465 g), MBAA (81 μ L of 2% solution), Irgacure 1173 (57 mg), and H₂O (10 g) in a glass bottle. After three degassing cycles, the bottle was sealed under N₂, the resulting solution was injected into mold was exposed to UV light (365 nm wavelength, 8W) for 1h to synthesize hydrogel.”

Nanoparticles	Lap-shear strength	Adhesion energy	Elastic modulus	Equilibrium swelling degree	Adhesion energy calculation
Silica Nanoparticles ¹		6-10 J/m ²	10 \pm 1.0 kPa	41	$G_{adh}=3(F/w)2/(2Eh)^a$
HAp nanoparticles ²	2-8 kPa				
PDA-nanoparticles-PVA ³	1 kPa		45 \pm 2 kPa	17	
Colloidal supraballs ⁴		10-100 J/m ²	9 kPa 18 kPa	3.85 2.46	$G_{adh}=3(F/w)2/(2Eh)^a$
Mesoporous nanoparticles ⁵		5-35 J/m ²	10 kPa		$G_{adh}=3(F/w)2/(2Eh)^a$
Activated silica nanoparticles, this article	~23 kPa	~400 J/m ²	21 \pm 8 kPa	620	$G_{adh}=F_p/w^b$

Annotation

a, F donate measured adhesive force, w and h donate the width and thickness of the sample, respectively.

b, F_p donate measured plateau force in the steady-state region of peeling process, w donate width of the sample.

Supplementary Table 1 The adhesive performance of ANP glue in adhering PDMA hydrogels is compared with previous reported nanoparticles-based glues.

References

- [1] Nature, 2014, 505, 382
- [2] Acta Biomater., 2017, 57, 404
- [3] Acta Biomater., 2022, 152,171
- [4] Chem. Mater., 2022, 34, 584
- [5] ACS Appl. Mater. Interfaces, 2017, 9, 31469

Reviewer #3 (Remarks to the Author):

The authors have carefully revised the manuscript and addressed all the concerns. The manuscript has been improved a lot, and it is suitable for publication now.

Response: We sincerely appreciate the reviewer's diligent evaluation and constructive feedback. We are grateful for their acknowledgement of our efforts in revising the manuscript to address all concerns.